# Synthetic HDL Nanoparticles Delivering Docetaxel and CpG for Chemoimmunotherapy of Colon Adenocarcinoma

**DOI:** 10.3390/ijms21051777

**Published:** 2020-03-05

**Authors:** Lindsay M. Scheetz, Minzhi Yu, Dan Li, María G. Castro, James J. Moon, Anna Schwendeman

**Affiliations:** 1Department of Pharmaceutical Sciences, College of Pharmacy, University of Michigan, Ann Arbor, MI 48109, USA; scheetzl@umich.edu (L.M.S.); minzhiyu@umich.edu (M.Y.); dansmile@umich.edu (D.L.); 2Biointerfaces Institute, University of Michigan, Ann Arbor, MI 48109, USA; 3Department of Neurosurgery, University of Michigan School of Medicine, Ann Arbor, MI 48109, USA; mariacas@umich.edu; 4Department of Cell and Developmental Biology, University of Michigan School of Medicine, Ann Arbor, MI 48109, USA; 5Department of Biomedical Engineering, University of Michigan, Ann Arbor, MI 48109, USA

**Keywords:** co-delivery, chemotherapy, immunotherapy, colon carcinoma

## Abstract

Colon carcinomas comprise over two-thirds of all colorectal cancers with an overall 5-year survival rate of 64%, which rapidly decreases to 14% when the cancer becomes metastatic. Depending on the stage of colon carcinoma at diagnosis, patients can undergo surgery to attempt complete tumor resection or move directly to chemotherapy with one or a combination of drugs. As with most cancers, colon carcinomas do not always respond to chemotherapies, so targeted therapies and immunotherapies have been developed to aid chemotherapy. We report the development of a local combination therapy for colon carcinoma whereby chemo- and immunotherapeutic entities are delivered intratumorally to maximize efficacy and minimize off-target side effects. A hydrophobic chemotherapeutic agent, docetaxel (DTX), and cholesterol-modified Toll-like receptor 9 (TLR9) agonist CpG (cho-CpG) oligonucleotide are co-loaded in synthetic HDL (sHDL) nanodiscs. In vivo survival analysis of MC-38 tumor-bearing mice treated intratumorally with DTX-sHDL/CpG (median survival; MS = 43 days) showed significant improvement in overall survival compared to mice treated with single agents, free DTX (MS = 23 days, *p* < 0.0001) or DTX-sHDL (MS = 28 days, *p* < 0.0001). Two of seven mice treated with DTX-sHDL/CpG experienced complete tumor regression. None of the mice experienced any systemic toxicity as indicated by body weight maintenance and normal serum enzyme and protein levels. In summary, we have demonstrated that chemo- and immunotherapies can be co-loaded into sHDLs, delivered locally to the tumor, and can be used to improve survival outcomes significantly compared to chemotherapy alone.

## 1. Introduction

Currently, the standard-of-care for patients with colon adenocarcinoma is surgery, radiation, chemotherapy, and possibly immunotherapy via immune checkpoint blockade. Though effective, both chemotherapy and immune checkpoint blockade agents are administered systemically and reach the tumor itself in very limited amounts [1]. Moreover, neither chemotherapy nor immune checkpoint blockade directly elicits immune memory against the tumor, which is crucial for preventing tumor recurrence. Thus, combining chemotherapy with an immunostimulatory agent that promotes immune cell interaction with tumor antigens is an exciting strategy that is readily translatable to the clinical arena and is beginning to be explored [2,3].

Several nanoparticles have been used for co-delivery of these two moieties, but none of them have been approved for clinical use. In the early 2000s, Weigel et al. published data showing the enhanced antitumor effects of cyclophosphamide and topotecan when combined with CpG in mouse models of embryonal rhabdomyosarcoma. Almost a decade later, Buhtoiarov et al. demonstrated that co-delivery of CpG with vincristine, cyclophosphamide, and doxorubicin in murine melanoma and neuroblastoma models significantly improved the antitumor effects of multidrug chemotherapy regimens alone. In 2015, Lollo et al. used lipid nanocapsules with a cationic chitosan shell to deliver paclitaxel and CpG simultaneously to glioblastoma tumors in mice through convection-enhanced delivery and showed that this co-delivery significantly improved survival outcomes compared to Taxol^®^ and separate injections of paclitaxel and CpG [4,5,6].

Synthetic high-density lipoprotein nanoparticles (sHDL) are very attractive nanoparticles for drug delivery applications that have been utilized for delivery of hydrophobic small molecule drugs [7], oligonucleotides [8], therapeutic proteins [9], and vaccine antigens [10,11,12,13]. They show increased uptake by cells expressing the endogenous HDL receptor scavenger receptor B1 (SR-B1) [8], long circulation times due to lack of recognition by the reticuloendothelial system, and safe elimination as the nanoparticles are biodegradable [8,14,15]. Moreover, sHDL nanoparticles have been extensively clinically tested for cardiovascular applications and have been proven safe and biocompatible for use in humans [16,17,18,19,20]. 

In the context of cancer, multi-functional sHDL nanoparticles carrying nucleic acids, antigens, and chemotherapeutic small molecules have shown significant tumor regression in B-cell lymphoma, ovarian cancer, and colorectal cancer mouse models [8,13,21,22,23,24,25,26]. One of the primary mechanisms by which sHDL nanoparticles are so effective is through their recognition by SR-B1 receptors, which are known to be overexpressed in most cancers [27,28]. This high receptor protein expression allows cancer cells to accumulate cholesterol through HDL particles to fuel cell metabolism and cancer cell proliferation [29]. Additionally, the small size of sHDLs of 8–14 nm allows more efficient trafficking throughout the tumor mass as compared to traditional liposomal nanoparticles of 100–300 nm [16,30]. 

Colon adenocarcinomas constitute the vast majority of all diagnosed colorectal cancer cases with an incidence rate of more than 140,000 people per year. Although the average 5-year survival rate for patients diagnosed with colorectal cancer is greater than 60%, colorectal cancer remains the second leading cause of death in the United States [31]. This is likely due to the fact that nearly half of treated patients develop recurrent disease [32]. Currently, the standard-of-care for patients with colon adenocarcinoma is surgery, radiation, chemotherapy, and possibly immunotherapy via immune checkpoint blockade. Though effective, both chemotherapy and immune checkpoint blockade agents are administered systemically and reach the tumor itself in a very limited amount. Moreover, neither chemotherapy nor immune checkpoint blockade directly elicits immune memory against the tumor, which is crucial for preventing tumor recurrence. Thus, combining chemotherapy with an immunostimulatory agent that promotes antigen-presenting cells’ interactions with tumor antigens is an exciting strategy that is currently being explored in clinical trials [2,3]. 

We hypothesize that sHDL encapsulating docetaxel chemotherapy and decorated with Toll-like receptor 9 (TLR9) agonist CpG oligonucleotide will ensure effective delivery to colon adenocarcinoma tumors, leading to suppressed tumor growth and long-term survival as compared to monotherapy delivery. TLR9 agonists interact with their receptors on a plethora of immune cells and are primarily involved in the activation and maturation of dendritic cells and the differentiation of B cells. These dendritic cells and B cells can then undergo cross-presentation of tumor-specific antigens and secretion of anti-tumor antibodies, respectively [33,34]. The source of these antigens is often apoptotic tumor cells, which can be directly produced by chemotherapeutic-mediated killing of tumor cells. We have previously reported the combination of CpG with chemotherapy for the treatment of glioblastoma [34] and, given the promising results of this study, we have decided to implement it in a different, yet aggressive, tumor model. Here, we describe the superior antitumor efficacy of co-delivering docetaxel chemotherapy and TLR9 agonist CpG on sHDL nanoparticles to tumors over docetaxel alone in the treatment of colon carcinoma. 

## 2. Results

### 2.1. Drug and Adjuvant Loading into sHDL do not Affect Size or Shape of sHDL

DTX is loaded in sHDL particles using a simple co-lyophilization/rehydration method as described previously [20,23]. DLS analysis showed that DTX-loaded sHDLs have an average particle size of 11.1 nm, which is similar to that of blank sHDL particles. Insertion of CpG-chol slightly increased the particle size to 11.3 nm (Figure 1A). All formulations demonstrated a high purity (>90%) in the GPC analysis (Figure 1B). The consistent size of DTX-loaded sHDL both before and after loading of DTX and CpG-chol in conjunction with high formulation purity was satisfactory, allowing us to proceed with in vitro and in vivo studies. 

### 2.2. SR-B1 Is Highly Expressed by MC38 Cells and Plays a Role in Cellular Uptake of sHDL 

To validate our choice of colon cancer model for evaluating our DTX-loaded sHDL formulations, we examined MC38 cells’ SR-B1 expression level to verify that sHDL could be recognized and endocytosed by MC38 cells. Western blot analysis of MC38 cell lysate showed high expression of SR-B1 relative to actin band density. This SR-B1 expression level was comparable to that of other tumor cell lines with known increased expression of SR-B1 and a favorable response to SR-B1 targeted therapeutic strategies, such as B16F10 and 4T1 (Figure 2A) [1,35,36]. We then examined sHDL uptake by MC38 cells in vitro (Figure 2B). When cultured with DiD-labeled sHDL for three hours, MC38 cells exhibited dose-dependent uptake of sHDL (Figure 2B,C). To confirm that SR-B1 played a role in this uptake, we inhibited SR-B1 on MC38 cells by adding Block lipid transporter 1 (BLT-1) to cell cultures for 1 h before treatment with DiD-sHDL again for 3 h. Indeed, we observed a marked decrease in cellular uptake of sHDL with increasing pretreatment doses of BLT-1 (*p* < 0.01) (Figure 2D). Together, these results show that MC38 cells exhibit high SR-B1 expression levels and efficient cellular uptake of sHDL. 

### 2.3. Delivery of DTX by sHDL Maintains the Cytotoxic Effect of DTX on MC38 Cells 

After validating the sHDL scavenging potential of MC38 cells, we tested the cytotoxic potential of DTX-loaded sHDL on MC38 cells in vitro. After 48 h of incubation with free DTX or DTX-loaded sHDL (DTX-sHDL), MC38 cells were analyzed by MTT assay using a UV spectrophotometer. We observed no significant differences in cytotoxicity at higher doses of DTX between the two treatment groups. DTX-sHDL induced cell death at a similar rate to DTX alone after just 48 h in doses of 16 and 24 micrograms per ~50,000 cells (Figure 2E). 

### 2.4. Combination of Immunostimulatory Agent with DTX Increases Antitumor Effects and Prolongs Survival

While we have demonstrated that sHDL enhances the delivery of DTX in vitro, single-agent therapy is often insufficient for total eradication of the tumor, especially in colon adenocarcinoma. Thus, we decided to incorporate immunostimulatory agent CpG1826 into our DTX-sHDL formulation to test the efficacy of combination therapy relative to single-agent chemotherapy and to determine whether this additional component would augment sHDL’s delivery enhancement of DTX in vivo. We observed significantly reduced tumor growth in mice treated with DTX-sHDL/CpG compared with DTX-sHDL or DTX alone, indicating the enhanced antitumor effect of combination therapy (Figure 3B). Mice treated with DTX-sHDL/CpG survived significantly longer (median survival = 43 days) (*p* < 0.0001) than mice treated with DTX-sHDL (median survival = 28 days) or DTX alone (median survival = 23 days) (Figure 3C). Two of seven mice treated with DTX-sHDL/CpG experienced complete tumor regression. None of the mice experienced any systemic toxicity as indicated by body weight maintenance and normal serum enzyme and protein levels (Figure 4). Overall, DTX-sHDL combined with the immunostimulatory agent CpG1826 significantly improved animal survival as compared with single agent chemotherapy via DTX-sHDL. 

## 3. Discussion

In this work, we hypothesized that sHDL encapsulating docetaxel chemotherapy would enhance the delivery of docetaxel to colon adenocarcinoma cells and that combining this single-agent therapy with TLR9 agonist CpG would augment the antitumor efficacy of the monotherapy, suppress tumor growth, and prolong animal survival. Through several in vitro experiments, we showed that sHDL uptake by MC-38 cells is indeed mediated by SR-B1, which is highly expressed on these cells. MC-38 cells exhibited a large capacity for sHDLs, as evidenced by quantitative flow cytometry-based identification of sHDL-positive cells and qualitative confocal microscopy images. These findings agree with the current literature reporting that high SR-B1 expression allows for greater accumulation of cholesterol in tumor cells via HDL uptake to prolong tumor cell survival [7,35]. Following incubation of MC-38 cells with either free DTX or DTX-sHDL, we observed that DTX-sHDL induced cell death at a similar rate to DTX at multiple dose levels. This result supports the hypothesis that sHDL can deliver DTX to tumor cells without compromising its cytotoxic activity; this potentiates the clinical translation of sHDL nanoparticles as drug delivery vehicles for hydrophobic chemotherapeutic molecules. We tested our formulation in vivo with an additional immunostimulatory component, CpG oligonucleotide. Indeed, CpG significantly improved the antitumor efficacy of DTX, suppressing tumor growth and prolonging survival in mice treated with DTX-sHDL/CpG as compared to mice treated with DTX-sHDL or DTX alone. Complete responses were achieved in two of the seven mice treated with DTX-sHDL/CpG, supporting the hypothesis that combination therapy with an immunostimulatory component would augment the antitumor efficacy of DTX alone.

Surprisingly, there were no significant differences in tumor growth or survival between the DTX-sHDL and DTX treatment groups. DTX-sHDL treatment marginally improved the animal survival rates as compared to DTX treatment, but no differences were observed in tumor growth rates. These findings are likely due to the intratumoral route of administration of DTX and DTX-sHDL, which does not showcase the tumor targeting abilities of sHDL nanoparticles as seen after intravenous administration [16,34,37]. Moreover, we only tested one dose of DTX, and it is possible that DTX-sHDL may be more effective than free DTX at lower doses. In 2015, Li et al. reported that diblock copolymer nanoparticle delivery of doxorubicin (DOX) slowed tumor growth more effectively than free DOX at a dose of 1 mg/kg [38]. In 2016, Jin et al. demonstrated that delivery of ursolic acid (UA) by chitosan nanoparticles increased the anti-angiogenic potency of UA tenfold compared to free UA administration [39]. Combinatorial chemotherapy studies by Camacho et al. also showed that liposomal delivery of both DOX and 5-fluorouracil (5FU) was able to regress tumors by more than 90% compared to 39% tumor regression by delivery of free DOX + 5FU [40]. Additionally, a deeper mechanistic understanding of this combination therapy is needed through delineating the individual therapeutic contributions of free DTX versus free CpG as well as evaluation of this platform in an orthotopic mouse model of colon adenocarcinoma. 

Several other nanoparticles have been used for the co-delivery of chemotherapeutic and immune stimulatory agents, but none of them have been approved for clinical use. In the early 2000s, Weigel et al. published data showing the enhanced antitumor effects of cyclophosphamide and topotecan when combined with CpG in mouse models of embryonal rhabdomyosarcoma. Almost a decade later, Buhtoiarov et al. demonstrated that co-delivery of CpG with vincristine, cyclophosphamide, and doxorubicin in murine melanoma and neuroblastoma models significantly improved the antitumor effects of multidrug chemotherapy regimens alone. In 2015, Lollo et al. used lipid nanocapsules with a cationic chitosan shell to deliver paclitaxel and CpG simultaneously to glioblastoma tumors in mice through convection-enhanced delivery and showed that this co-delivery significantly improved survival outcomes compared to Taxol^®^ and separate injections of paclitaxel and CpG [4,5,6]. 

Here, we have described the superior antitumor efficacy of co-delivering docetaxel chemotherapy and TLR9 agonist CpG on sHDL nanoparticles to MC-38 colon adenocarcinoma tumors when compared to docetaxel alone. Our findings are significant because sHDL has already been demonstrated to be safe and well-tolerated in high doses in the clinic, making it a readily translatable platform for co-delivered combination therapy for colon adenocarcinoma patients. Indeed, no systemic toxicity was elicited by the sHDL-DTX-CpG treatment. We have also shown that this platform is translatable to other cancers as it has been previously effective in a glioblastoma model [34]. Our results give promise to the evolution of highly effective, minimally invasive, and non-toxic multi-functional cancer therapies for colon adenocarcinoma.

## 4. Materials & Methods

### 4.1. Materials

Egg sphingomyelin (eSM) was acquired from Avanti Polar Lipids. Apolipoprotein A-1 mimetic peptide 22A was acquired from Genscript Inc., Piscataway, NJ, USA. Docetaxel (DTX) was acquired from Cayman Chemicals., Ann Arbor, MI, USA. MC-38 cells were purchased from Kerafast, Inc., Boston, MA, USA. Cholesterol-modified CpG1826 was custom ordered from Integrated DNA Technologies, Coralville, IA, USA. 

### 4.2. Formulation and Characterization of DTX-sHDL

The docetaxel-loaded sHDL (DTX-sHDL) was prepared as described previously [34]. Briefly, 22A, egg sphingomyelin (eSM), and DTX were dissolved in acetic acid. The acetic acid solutions of 22A, eSM, and DTX were mixed and freeze-dried for 24 h (mass ratio of 22A: eSM: DTX = 1:2:0.05). The lyophilized powder was rehydrated by PBS (pH 7.4). Three heat-cooling cycles (50 °C, 5 min followed by room temperature for 5 min) were performed to form DTX-sHDL. DTX-sHDL/CpG particles were prepared by incubating DTX-sHDL with CpG-Cholesterol in 10 mM phosphate buffer at room temperature for 2 h. The particle size of DTX-sHDL was analyzed by dynamic laser scattering (DLS). The purity of the DTX-sHDL nanoparticles was evaluated by gel permeation chromatography (GPC) at 220 nm using Tosoh TSK gel G3000SWx 7.8 mm × 30 cm column (Tosoh Bioscience, King of Prussia, PA, USA). 

### 4.3. In Vitro Uptake Assays

MC-38 cells were cultured in RPMI medium supplemented with 10% Fetal Bovine Serum and 1% Penicillin/Streptomycin antibiotics. When cells reached their exponential growth phase, they were trypsinized and plated on 12-well tissue culture plates at 50,000 cells per well to be incubated overnight at 37 °C to allow adherence. HDL was labeled with lipophilic dye DiD at a ratio of 2:1:0.01 eSM:22A:DiD. DiD-labeled sHDL was passed through a desalting column (MWCO 7kDa) to remove free dye molecules prior to use. 

Cells were dosed with three different concentrations of DiD-sHDL normalized by 22A concentration to evaluate the effect of the dose on cell uptake. Following dosing, cells were incubated at 37 °C for 3 h and then washed with PBS before analysis by confocal microscopy on a Nikon A1si confocal microscope or by FACS on a CytoFlex cytometer. FlowJo and ImageJ were used for quantitative analysis. 

For the block lipid transport-1 (BLT-1) inhibition experiment, cells were pretreated with SR-B1 inhibitor BLT-1 with different concentrations for 1 h. Then, DiD-sHDL was added to each well (final 22A concentration = 10 mcg/mL). The cells were further incubated for 3 h followed by FACS analysis.

### 4.4. In Vitro Cytotoxicity Assay

MC-38 cells were cultured in RPMI medium supplemented with 10% Fetal Bovine Serum and 1% Penicillin/Streptomycin antibiotics. When cells reached their exponential growth phase, they were trypsinized and plated on 96-well tissue culture plates at 10,000 cells per well to be incubated overnight at 37 °C to allow adherence. Cells were dosed with six different doses of DTX in either free drug form or encapsulated in HDL to test the effect of increasing dose on cell death. Following dosing, cells were incubated at 37 °C for 48 h before analysis using the CellTiter 96^®^ AQ_ueous_ Non-Radioactive Cell Proliferation Assay (MTS) from Promega, Madison, WI, USA. Absorbance at 490 nm was quantified using a BioTek SynergyNEO spectrophotometer. Negative control wells (without treatment) were considered to have the maximum absorbance at 100% viability, and the viability of other wells was calculated as the ratio of treated well absorbance to untreated well absorbance. 

### 4.5. Western Blot of SR-B1 Expression on Murine Cancer Cells

Four different murine cancer cell lines–MC-38, B16-F10, CT-26, and 4T1–were cultured, trypsinized, and spun down so that cell pellets could be collected and flash frozen. Cell lysates were prepared and centrifuged at 16,000× *g* for 20 min at 4 °C. Supernatant was collected and stored on ice to perform total protein quantification by BCA assay. Samples were normalized to 30 µg total protein for loading onto an SDS-PAGE gel. The gel was run, transferred, and incubated with SR-B1 and actin primary antibodies overnight at 4 °C followed by incubation with HRP-conjugated secondary antibody at room temperature. The gel was imaged using a BioRad chemiluminescent imager and analyzed with ImageJ. 

### 4.6. In Vivo Treatment Using Combination Chemotherapy and Immunotherapy

Thirty-five female C57BL/6 mice aged 7–8 weeks (Charles River Laboratories, Wilmington, MA, USA) were inoculated with 1 million MC-38 cells at a concentration of 10 million cells/mL subcutaneously superior to the right flank. On day 8 after tumor inoculation, mice were split into four groups of seven for treatments. Mice were injected intratumorally with (1) PBS, (2) DTX, (3) DTX-HDL, or (4) DTX-HDL/CpG twice a week at 1 mg/kg DTX and 15 µg CpG for five treatments. Mice were euthanized when tumors surpassed 15 mm in one dimension or ulcerated extensively. Blood samples were taken four days after the final treatments were administered for serum isolation, and analysis of toxicity markers was performed by the In-Vivo Animal Core Animal Diagnostic Laboratory. 

## Figures and Tables

**Figure 1 ijms-21-01777-f001:**
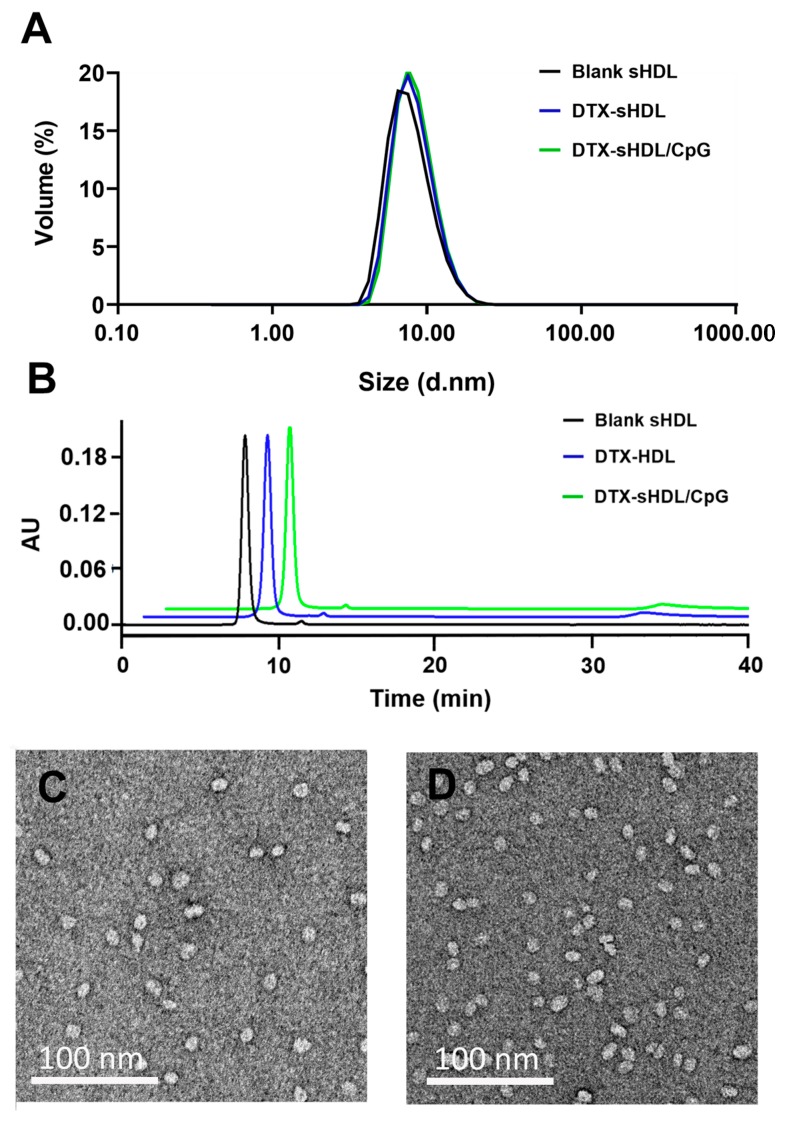
Characterization of sHDL loaded with DTX and CpG. (**A**) Dynamic light scattering (DLS) analysis of blank sHDL (black curve), DTX-sHDL (blue curve), DTX-sHDL/CpG (green curve). (**B**) Gel permeation chromatography (GPC) analysis of blank sHDL (black curve), DTX-sHDL (blue curve), DTX-sHDL/CpG (green curve). (**C**,**D**) Transmission electron microscopy (TEM) images of DTX-sHDL (**C**) and DTX-sHDL/CpG (**D**) particles.

**Figure 2 ijms-21-01777-f002:**
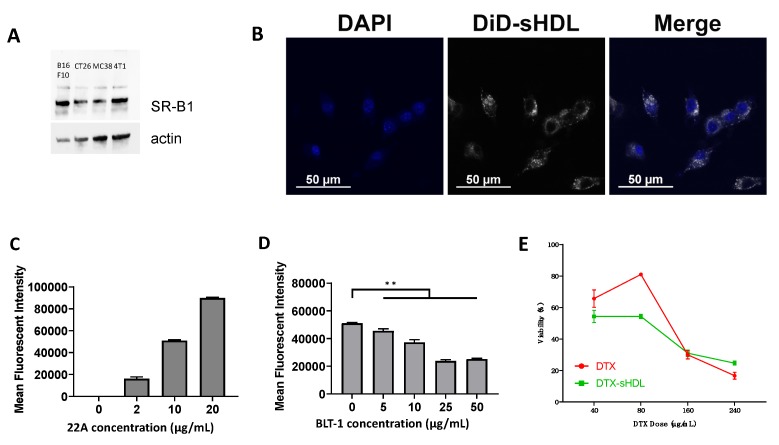
Cell uptake of sHDL by MC38 cells. (**A**) Western blot analysis of SR-B1 expression by four cancer cell lines: B16F10, CT26, MC38, 4T1. (**B**) Confocal microscope images of DiD-labeled sHDL by MC38 cells counterstained with DAPI at a 22A concentration of 20 μg/mL. (**C**) Quantitative analysis of DiD-labeled sHDL uptake by MC38 cells. (**D**) Quantitative analysis of DiD-labeled sHDL uptake by MC38 cells when pre-incubated with the SR-B1 blocking molecule BLT-1. (**E**) Cytotoxicity analysis of MC38 cells incubated for 48 h in a 96-well plate with either free DTX or DTX-sHDL at different drug molecule concentrations (** *p* < 0.01).

**Figure 3 ijms-21-01777-f003:**
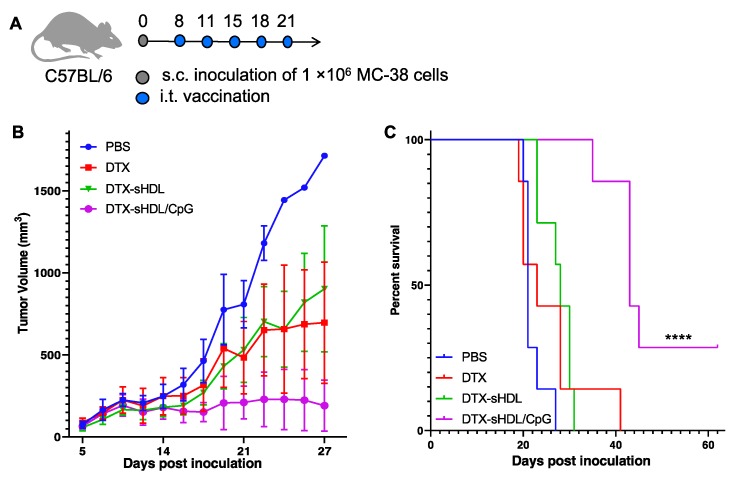
(**A**) Timeline of therapeutic animal study. (**B**) Tumor growth curves for mice treated with PBS (blue), DTX (red), DTX-sHDL (green), and DTX-sHDL/CpG (purple). (**C**) Kaplan–Meier survival curves for mice treated with PBS (blue), DTX (red), DTX-sHDL (green), and DTX-sHDL/CpG (purple) (**** *p* < 0.0001).

**Figure 4 ijms-21-01777-f004:**
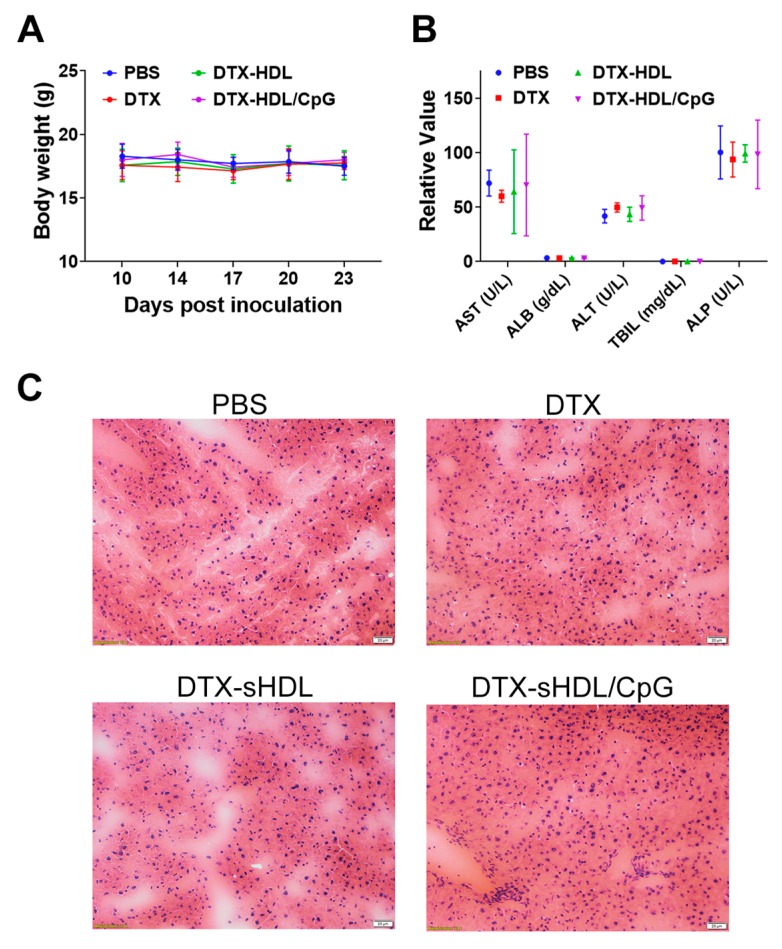
(**A**) Body weight measurements for study duration. (**B**) Liver panel toxicity analysis of aspartate aminotransferase (AST), albumin (ALB), alanine aminotransferase (ALT), total bilirubin (TBIL), and alkaline phosphatase (ALP). (**C**) H&E staining of livers from mice in each treatment group. Scale bars represent 20 microns.

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
