# Peer review of "Synthetic HDL Nanoparticles Delivering Docetaxel and CpG for Chemoimmunotherapy of Colon Adenocarcinoma"

_ijms, 2020, doi:10.3390/ijms21051777_

Round 1

Reviewer 1 Report

The study by Scheetz et al. demonstrates interesting results, and provides conclusions supported by the results.

Only few concerns should be addressed:

1. The Authors state that "SR-B1 is highly expressed by MC38 cells". Looking at the Fig. 2a it is clearly visible that all 4 tested cell lines highly express SR-B1. In addition, why melanoma cell line was used?

2. "DTX is loaded in sHDL particles using a simple co-lyophilization/rehydration method as described previously" needs reference.

3. The results could be discussed in the light on more recent papers.

Author Response

Reviewer 1:

The study by Scheetz et al. demonstrates interesting results, and provides conclusions supported by the results.

Only few concerns should be addressed:

The Authors state that "SR-B1 is highly expressed by MC38 cells". Looking at the Fig. 2a it is clearly visible that all 4 tested cell lines highly express SR-B1. In addition, why melanoma cell line was used?

Thank you for your comment. We tested multiple cell lines for which we had established animal models. The melanoma cell line was used for comparison because melanomas have been previously shown to express high levels of SR-B1 (Zamanian-Daryoush et al. Oncotarget, 2017).

"DTX is loaded in sHDL particles using a simple co-lyophilization/rehydration method as described previously" needs reference.

Thank you for your comment. We have inserted references.

The results could be discussed in the light on more recent papers.

Thank you for your comment. We have added in more discussion of recent publications.

Reviewer 2 Report

The manuscript entitled “Synthetic HDL nanoparticles delivering docetaxel and CpG for chemo-immunotherapy of colon adenocarcinoma” submitted by Lindsay M. Scheetz et al. describes the evaluation of anti-tumor activity of synthetic HDL nanoparticles loaded with docetaxel and CpG (an agonist of TLR9). Even this manuscript describes helpful results, it still some weakness before its acceptance for publication in the International Journal of Molecular Sciences.

It needs some major revisions listed below:

1-         The abstract is too long. The authors should shorten the text.

2-         The Figure 1B, 1C and 1D are not in the right orientation, the magnification of the images of fig 1C and fig 1D is not enough to observe clearly the size and morphology of nanoparticles.

3-         In Figure 2A, there is a protein band between SR-B1 and actin which is more pronounced in B16F10 cells. The authors should discuss this point. The Figures 2B and 2C are not informative, the authors should improve this experiment. They need to use other fluorescent markers such as DAPI, plasma membrane marker, etc…  In their text (page 9), there is some inconsistency: “uptake of sHDL (Figure 2B, C)”, “doses of BLT-1 (p<0.01) (Figure 2D)”. The results of Figure 2F are not commented in the text.

4-         The images presented in Figure 4C are blurring. The authors should improve these images

From all of these points, I consider that this manuscript is not suitable for publication.

Author Response

Reviewer 2:

The manuscript entitled “Synthetic HDL nanoparticles delivering docetaxel and CpG for chemo-immunotherapy of colon adenocarcinoma” submitted by Lindsay M. Scheetz et al. describes the evaluation of anti-tumor activity of synthetic HDL nanoparticles loaded with docetaxel and CpG (an agonist of TLR9). Even this manuscript describes helpful results, it still some weakness before its acceptance for publication in the International Journal of Molecular Sciences.

It needs some major revisions listed below:

The abstract is too long. The authors should shorten the text.

Thank you for your comment. We have shortened the abstract.

The Figure 1B, 1C and 1D are not in the right orientation, the magnification of the images of fig 1C and fig 1D is not enough to observe clearly the size and morphology of nanoparticles.

Thank you for your comment. We have modified Figure 1.

3-         In Figure 2A, there is a protein band between SR-B1 and actin which is more pronounced in B16F10 cells. The authors should discuss this point. The Figures 2B and 2C are not informative, the authors should improve this experiment. They need to use other fluorescent markers such as DAPI, plasma membrane marker, etc…  In their text (page 9), there is some inconsistency: “uptake of sHDL (Figure 2B, C)”, “doses of BLT-1 (p<0.01) (Figure 2D)”. The results of Figure 2F are not commented in the text.

            Thank you for your comments. In Figure 2, we compared the SR-B1 expression on MC38 with that on cells with known SR-B1 overexpression including B16F10 and 4T1. Results showed MC38 cells have similar levels of intact SR-B1 as other cell lines, which justified the use of SR-B1 targeting sHDLs particle. We have adjusted Figure 2 in A) to exclude the SR-B1 fragment band because it is irrelevant to our studies. and also in B) to DAPI staining with the fluorescence of DiD-sHDL. We have also adjusted the text to match Figure 2.

4-         The images presented in Figure 4C are blurring. The authors should improve these images

From all of these points, I consider that this manuscript is not suitable for publication.

            Thank you for your comment. We have increased the size and resolution of the images in 4C.

Round 2

Reviewer 2 Report

I appreciate the improvements made by the authors but there are still some points to which the authors have not responded with satisfaction. This is Figure 2, I am not convinced of the changes to this figure. On the one hand, it is not a question of eliminating the degradation product but of repeating the experiment by taking the precautions of not having degradation products by the proteases in order to be able to affirm which cell lines overexpress the protein SR -B1. On the other hand, the image of confocal microscopy is not informative enough. the magnification is too low to be able to appreciate the internalization of the sHDL nanoparticles.

For these reasons, I cannot accept this manuscript without answering these questions

Author Response

We thank the reviewer for their comments. We have repeated the Western blot and the cell uptake experiments shown in Figure 2. To avoid protein degradation in cell lysates used in the Western blot, we used a brand new protease inhibitor and took extra care during sample prep. We no longer see the band in question imaged on the first Western blot submitted but do see a faint band below the major SR-B1 band, indicating detection of a protein isoform with the polyclonal primary antibody used. For better visualization of nanodisc uptake in cells, we used a higher magnification on the confocal microscope and can now see punctate patterns within the MC-38 cells. 

Round 3

Reviewer 2 Report

Dear Editor,

Dear publisher,The authors have now answered the questions asked and have improved the manuscript. For these reasons, I accept the publication of this manuscript in International Journal of Molecular Science (IJMS).

Best regards

Nadir Bettache